# Establishment of an I-ELISA method based on multi-epitope fusion protein for diagnosis of human brucellosis

**Yujia Xie[1]☉, Liping Guo[1]☉, Xinru Qi[1], Shiqi Zhao[1], Qichuan Pei[1], Yixiao Chen[1], Qi Wu[1], Meixue Yao[1]\*, Dehui Yin**  **[1,2]\***

**1** Jiangsu Engineering Research Center of Biological Data Mining and Healthcare Transformation, Xuzhou Medical University, Xuzhou, Jiangsu, China, **2** Center for Medical Statistics and Data Analysis, School of Public Health, Xuzhou Medical University, Xuzhou, Jiangsu, China

☉ These authors contributed equally to this work
\* yaomeixue@163.com (MY); yindh16@xzhmu.edu.cn (DY)

## Abstract

### Background

Brucellosis is a significant zoonotic disease that impacts people globally, and its diagnosis has long posed challenges. This study aimed to explore the application value of multi-epitope fusion protein in the diagnosis of human brucellosis.

### Methods

Eight important *Brucella* outer membrane proteins (OMPs) were selected: BP26, omp10, omp16, omp25, omp2a, omp2b, and omp31. Bioinformatics techniques were used to predict the immune epitopes of these proteins, and a multi-epitope fusion protein was designed. This fusion protein was used as the antigen for indirect enzyme-linked immunosorbent assay (iELISA) testing on 100 positive and 96 negative serum samples. The performance of the fusion protein in diagnosing brucellosis was evaluated using receiver operating characteristic (ROC) curves.

### Results

A total of 31 epitopes were predicted from the eight proteins, and a multi-epitope fusion protein was successfully obtained. For the detection of human serum samples, the area under the ROC curve (AUC) of the fusion protein was 0.9594, with a positive diagnostic accuracy of 91.26% and a negative diagnostic accuracy of 93.55%. The area under the ROC curve (AUC) for lipopolysaccharides (LPS) was 0.9999, with a positive diagnostic accuracy of 100% and a negative diagnostic accuracy of 98.97%.

### Conclusions

The fusion protein constructed using bioinformatics techniques, as the diagnostic antigen, showed significantly reduced cross-reactivity and enhanced specificity, improving

**Data availability statement:** The datasets supporting the conclusions of this article are included within the article and its Supporting information file.

**Funding:** This work was supported by Xuzhou Science and Technology Bureau (Grant number KC23306), the Medical Research Program of Jiangsu Commission of Health (Grant number Z2023080), Postgraduate Research & Practice Innovation Program of Jiangsu Province (Grant number KYCX23-2963), and QingLan Project of Jiangsu Province (2024). The funders had no role in study design, data collection and analysis, decision to publish, or preparation of the manuscript.

**Competing interests:** The authors have declared that no competing interests exist.

diagnostic accuracy. This not only saves time but also avoids the preparation of LPS antigens, making the diagnostic process safer and more convenient.

## Author summary

Brucellosis is a serious zoonotic disease affecting people worldwide, and its diagnosis has always been challenging. Traditional diagnostic methods typically rely on lipopolysaccharides (LPS) antigen, which are not only complex and costly to prepare but also may present issues with cross-reactivity. In this study, we utilized bioinformatics techniques to predict the immunogenic epitopes of *Brucella* outer membrane proteins and designed a multi-epitope fusion protein. By conducting indirect enzyme-linked immunosorbent assay (iELISA) on 100 positive and 96 negative serum samples, we evaluated the performance of this fusion protein in the diagnosis of brucellosis and found that it significantly outperforms traditional LPS antigen in reducing cross-reactivity. This not only saves time but also avoids the preparation of LPS antigen, making the diagnostic process safer and more convenient.

## 1. Introduction

*Brucella* is a Gram-negative, facultative anaerobic bacterium that causes zoonotic, hypersensitivity infectious diseases. Human brucellosis cases have been reported in over 170 countries and regions worldwide, with approximately 500,000 new cases each year [1]. The situation in China is concerning, with the number of cases increasing annually, surpassing 70,000 in 2023. The range of susceptible animals is broad, including ruminants (sheep, cattle, deer), pigs, dogs, horses, camels, and rodents. In livestock, brucellosis can cause late-term abortions [2], intrauterine fetal death, premature births, and reduced fertility in offspring. Humans are also susceptible, with transmission primarily occurring through skin mucosa, the digestive tract, and the respiratory tract [3]. Patients may experience recurrent high fever, sweating, weakness, chills, myalgia, joint pain, and weight loss, with possible progression to arthritis. In severe cases, this can lead to loss of work ability, posing a serious threat to human life and health [4,5]. Due to the impact of brucellosis on various systems and organs and its nonspecific symptoms, it is often misdiagnosed as other systemic diseases [6,7]. Therefore, establishing accurate diagnostic methods is crucial for the prevention and control of brucellosis.

The diagnostic methods for *Brucella* primarily include bacteriological, molecular, and serological diagnostics. Isolating and culturing *Brucella* is the fundamental method for diagnosing brucellosis and is considered the "gold standard" for detection. However, this method has a low detection rate, is complex to perform, requires stringent laboratory conditions, and poses safety risks [8]. Molecular biological techniques, such as PCR and loop-mediated isothermal amplification (LAMP), offer advantages like high sensitivity, strong specificity, high safety, and repeatability. The entire reaction takes place under constant temperature conditions and requires relatively short processing times. However, molecular biological methods are susceptible to post-amplification contamination and false-positive results [9,10]. Serological diagnostics primarily include agglutination tests and enzyme-linked immunosorbent assays (ELISA). The sensitivity and specificity of tube agglutination tests are lower compared to other methods. ELISA possesses advantages such as high sensitivity, strong specificity, and requiring a small sample volume, making it suitable for screening large numbers of samples as well as serving as a confirmatory test for brucellosis [11–13]. However, traditional antigens used in

serological diagnosis, such as lipopolysaccharides (LPS) and whole-cell antigens, are prone to cross-reactivity with *Escherichia coli* O157:H7 and *Yersinia enterocolitica* O9. Therefore, the search for new diagnostic antigens is of great significance for improving the serological diagnosis of brucellosis [14].

Based on the rapid development of genetic engineering, genes encoding functional proteins are purposefully linked together to express a desired protein. Proteins obtained by fusing different gene coding regions under artificial conditions are referred to as fusion proteins (FP). Using fusion protein technology, novel target proteins with multiple functions can be constructed and expressed [15]. Currently, researchers both domestically and internationally have identified many immunogenic proteins of *Brucella* that can be used for serological diagnosis and subunit vaccine development [5,16]. Among these, outer membrane proteins have been shown to have good immunogenicity and protective effects under laboratory conditions, such as OMP10, OMP16, OMP25, OMP31, OMP2a, OMP2b, and BP26 [17].

Bioinformatics analysis is an emerging science that combines life sciences, computer science, and mathematics to accelerate vaccine design, disease diagnosis, treatment, and drug screening processes [18,19]. Since B-cell epitopes are the molecular sites recognized by antibodies [20], various online prediction tools for B-cell epitopes have been developed with the advancement of bioinformatics technology. The Immune Epitope Database (IEDB) provides a comprehensive immune epitope database, including linear and structural epitopes, with extensive data from different pathogens, vaccines, and disease types, offering reliable information. Artificial neural network-based B-cell epitope prediction server (ABCpred) uses advanced machine learning algorithms to accurately identify potential immunogenic sites. B-cell epitope prediction server (BCpreds) optimizes its model with a large amount of experimental data, enhancing the reliability of its predictions. Support Vector Machine (SVM) combines the Tri-peptide similarity and Propensity scores (SVMTriP) with other machine learning methods to provide high prediction accuracy, effectively identifying and predicting the three-dimensional structural epitopes of antigens. Unlike prediction tools based solely on linear sequences, SVMTriP considers the spatial structure information of proteins, improving the accuracy of predictions. This study employed these prediction methods to complement each other, ensuring a comprehensive analysis and high-precision prediction of B-cell epitopes. This approach ensures that the outer membrane protein fusion protein not only includes high-prediction-value epitopes but also integrates experimentally validated effective epitopes, enhancing the immunogenicity and reliability of the fusion protein.

This study used bioinformatics-related technologies to predict B-cell antigenic epitopes on *Brucella* outer membrane proteins. The predicted epitopes were linked together to construct a fusion protein amino acid sequence capable of specifically recognizing *Brucella* infection antibodies. The fusion protein was then purified and expressed. A new *Brucella* multi-epitope fusion protein was designed, and an indirect enzyme-linked immunosorbent assay (iELISA) was developed using this protein to address the deficiencies in current *Brucella* disease immunoassay methods.

## 2. Materials and methods

### 2.1. Serum samples

A total of 100 *Brucella*-positive serum samples confirmed by standard agglutination test (SAT) and 96 negative serum samples were provided by the Xuzhou Center for Disease Control and Prevention. All samples were collected between January 2022 and November 2023 and tested by a commercial SAT kit (231116, TSINGTAO SINOVA-HK BIOTECHNOLOGY, Qingdao, China). Additionally, 40 serum samples (Provided by the First Affiliated Hospital of Jilin

University and collected in 2020) from patients infected with bacteria other than *Brucella* were acquired in the laboratory for cross-reactivity validation.

## 2.2. Preparation of fusion proteins

**2.2.1. Prediction of B-cell linear epitopes. Selection of major *Brucella* outer membrane proteins:** By reviewing relevant literature on *Brucella* outer membrane proteins in the PubMed database, we selected outer membrane proteins that were commonly found in *B. ovis*, *B. abortus*, and *B. suis*. We prioritized proteins that were conserved in these three species of *Brucella*. After identifying the target outer membrane proteins, we queried the amino acid sequences of these proteins in the NCBI Protein database (http://www.ncbi.nlm.nih.gov/protein/). Subsequently, we employed the BLAST tool within Protein Tools to compare the amino acid sequences of *Brucella* with those of outer membrane proteins from other bacterial species. Our analysis specifically aimed to identify conserved amino acid sequences across the three aforementioned *Brucella* species, thereby enhancing the specificity of our findings.

**Prediction of B-cell epitopes for *Brucella* proteins:** After obtaining the amino acid sequences of *Brucella* proteins, four B-cell epitope prediction tools were used to enhance the accuracy of epitope prediction [21–24]. These tools include ABCpred (https://webs.iiitd.edu.in/raghava/abcpred/index.html, default threshold value: 0.5), SVMTriP (http://sysbio.unl.edu/SVMTriP, no default threshold value), BCPreds (http://ailab-projects2.ist.psu.edu/bcpred/predict.html, default threshold value: 0.75), and Bepipred Linear Epitope Prediction 2.0 (http://tools.iedb.org/bcell/, default threshold value: 0.5). We used these default threshold values for our analysis, and scores of epitope prediction above these thresholds were considered as possible B-cell epitopes.

Once the epitopes were predicted, they were further integrated with B-cell epitopes that had been experimentally validated in the IEDB database. This approach ensures that the outer membrane protein fusion proteins not only contain high-prediction-score epitopes but also include experimentally validated effective epitopes, thereby enhancing the immunogenicity and reliability of the fusion proteins.

**2.2.2. Construction of fusion protein amino acid sequences.** The predicted antigenic epitopes were concatenated with a linker "GGGS" in between to form the multi-epitope fusion protein amino acid sequence. The constructed amino acid sequence was then reverse-translated into codons and optimized for codon usage to be suitable for expression in *E. coli*. A His tag was added at the 3' end to facilitate purification. Finally, the gene encoding the *Brucella* multi-epitope fusion protein was synthesized by a biotechnology company (Beijing BGI Protein Research Center), and a fusion protein expression plasmid (pET30a) was constructed. The fusion protein was then prepared by prokaryotic expression (Details of the protocol for preparation are shown in the S1 Text).

**2.2.3. Prokaryotic expression of fusion protein. Fusion protein purification:** Purification of the fusion protein was performed using nickel affinity chromatography (Ni Sepharose 6 Fast Flow, GE Healthcare). Using a low-pressure chromatography system, the supernatant was loaded onto a Ni-IDA-Sepharose CL-6B affinity column pre-equilibrated with Ni-IDA Binding Buffer at a flow rate of 0.5 mL/min. The column was washed with Ni-IDA Binding Buffer at 0.5 mL/min until the $OD_{280}$ of the eluate reached baseline. The column was then washed with Ni-IDA Washing Buffer (20 mM Tris-HCl, 20 mM imidazole, 0.15 M NaCl, 8 M urea, pH 8.0) at a flow rate of 1 mL/min until the $OD_{280}$ of the eluate reached baseline. The target protein was eluted with Ni-IDA Elution Buffer (20 mM Tris-HCl, 250 mM imidazole, 0.15 M NaCl, 8 M urea, pH 8.0) at 1 mL/min, and the eluate was collected. The collected protein solution was added to a dialysis bag and dialyzed overnight against refolding buffer. After refolding, the solution was dialyzed against PBS for storage. SDS-PAGE analysis (12%) was performed to assess the purity of the protein.

## 2.3. Establishment and evaluation of the indirect ELISA method

Purified *Brucella* multi-epitope fusion protein was diluted to 10 μg/mL and 100 μL of this solution was used to coat each well of a 96-well microplate (Corning, USA). The plate was incubated overnight at 4°C. Afterward, the plate was washed three times with PBST (Phosphate Buffered Saline with Tween 20), 3 minutes each time, and then dried. Blocking was performed with 300 μL of blocking buffer (5% skim milk) per well at 37°C for 2 hours. The plate was washed again with PBST, and then 100 μL of 1:400 diluted human serum antibodies were added to each well, followed by incubation at 37°C for 1 hour. After washing three times with PBST, 100 μL of HRP-conjugated rabbit anti-human IgG (1:10,000 dilution, Thermo Fisher, USA) was added to each well and incubated at 37°C for 1 hour, followed by washing as before. The plate was then developed with 100 μL of TMB substrate solution at room temperature in the dark for 10 minutes. The reaction was stopped by adding 50 μL of stop solution (2 M $H_2SO_4$) to each well, and the $OD_{450}$ value was read using a microplate reader (Versa Max Microplate Reader, MD, USA). LPS was used as a control antigen (LPS was extracted from *B. abortus*, provided by the China Animal Health and Epidemiology Center, 3 mg/mL), and the same conditions were applied to test the above serum samples in triplicate wells. The average $OD_{450}$ values of each sample (All data are detailed in S1 Table) were used for ROC curve analysis to determine the sensitivity, specificity, and other metrics of the indirect ELISA method based Youden index.

## 2.4. Evaluation of cross-reactivity

In addition, the constructed iELISA was used to test the sera of other pathogen-infected individuals to evaluate cross-reactivity, with the detection steps being the same as described. Cross-reactivity was assessed by evaluating the $OD_{450}$ values against the cut-off values determined from the ROC curve analysis.

## 2.5. Statistical methods

The OD450 values were repeated 3 times for each sample, the average OD450 value for each sample is then calculated and used for ROC curve analysis. Data were statistically analyzed using IBM SPSS 24.0 software. Independent sample t-tests were conducted to assess the differences between positive and negative samples. A *P*-value of less than 0.05 was considered to indicate a significant difference between the two groups. Additionally, ROC curve analysis and scatter plot generation were performed using GraphPad Prism 9.5.0 software. Data sets of ROC curve analysis as follows: Control values, Average OD450 values of brucellosis Negative samples including 40 serum samples from patients infected with non-*Brucella* bacteria; Patient values, Average OD450 values of brucellosis Positive samples.

## 2.6. Ethical approval statement

The studies were reviewed and approved by the Ethics Committee of Xuzhou Medical University (approved number: xzhmu-20240201). No formal consent was obtained for anonymity.

## 3. Results

### 3.1. B-cell epitope prediction

Eight specific outer membrane proteins were selected as candidate targets. A total of 31 epitopes were predicted, as shown in Table 1. These epitopes were then concatenated to form the amino acid sequence of the fusion protein.

**Table 1. B-cell epitope prediction.**

| Protein | Epitope (amino acid sequence) | Start-end position | Length |
|---|---|---|---|
| BP26 | NQMTTQPARIAV | 31-42 | 10 |
| BP26 | QPIYVYPDDKNNLKEPTIT | 104-122 | 19 |
| BP26 | RPPMPMPIARG | 206-216 | 11 |
| BP26 | APDNSVPIAAGENSYNVSVVNVVFE | 225-248 | 24 |
| Omp10 | NLDNVSPPPPPAPVNAVPASTV | 28-49 | 22 |
| Omp10 | KGNLDSPTQFPNAPSTDMSAQSGTQV | 51-76 | 16 |
| Omp10 | AFAPDLTPG | 82-90 | 9 |
| Omp10 | QTKYGQGY | 111-118 | 8 |
| Omp10 | QGRFDGQTTGG | 160-170 | 11 |
| Omp16 | NLPNNAGDLGL | 30-40 | 11 |
| Omp25 | QPPVPAPVEV | 30-39 | 10 |
| Omp25 | TSTVGSIKP | 63-71 | 9 |
| Omp25 | NGLDDES | 147-153 | 7 |
| Omp2a | NNSRHDGQYGDFSDDRDVADGGVS | 116-139 | 24 |
| Omp2a | GGEDVDND | 208-215 | 8 |
| Omp2a | SSAATPNQNYGQWG | 274-287 | 14 |
| Omp2a | TKFGGEWKDTV | 341-351 | 11 |
| Omp2b | KGGDDVYSGTDRNGWD | 81-96 | 16 |
| Omp2b | NNSGVDGKYGNETSSG | 127-143 | 17 |
| Omp2b | NDGGYTGTTNYHI | 214-226 | 13 |
| Omp2b | PDQNYGQWGG | 287-296 | 10 |
| Omp2b | VSYIKFGGEWKNTVAEDN | 346-363 | 18 |
| Omp31 | VSEPSAPTAAP | 24-34 | 11 |
| Omp31 | FDKEDNEQVS | 63-72 | 10 |
| Omp31 | QAGYNWQLDNGVVLGA | 87-102 | 16 |
| Omp31 | GDDASALHMW | 168-177 | 10 |
| Omp2b Porin | VIEEWAAKVRGDVNITDQFSVWLQGAYSSAATPDQNYGQWG | 255-295 | 41 |

## 3.2. Preparation of the fusion protein

After prokaryotic expression, the target protein was detected in the supernatant. The purity of the purified fusion protein reached 90.1%. The results are shown in Fig 1.

## 3.3. Evaluation of the effect of I-ELISA for detecting *Brucella*

The sera of the four groups were analyzed and compared by unpaired t-test and the differences were statistically significant (Table 2). According to ROC curve analysis, the area under the diagnostic curve (AUC) for the fusion protein was 0.9594 (95% CI, 0.9313 - 0.9874), while the AUC for LPS was 0.9999 (95% CI, 0.9995 - 1.000), indicating both have good diagnostic value. Based on the Youden index, the cut-off value for the fusion protein diagnosis was 0.2774. At this value, the sensitivity of the method was 0.9063 (95% CI, 0.8313 - 0.9499), and specificity was 0.9400 (95% CI, 0.8752 - 0.9722). The cut-off value for diagnosing LPS was 0.1953, with a sensitivity of 1.0000 (95% CI, 0.9615 - 1.0000) and specificity of 0.9900 (95% CI, 0.9455 - 0.9995). The results are shown in Fig 2 and Table 3.

## 3.4. Evaluation of cross-reactivity

Using iELISA and based on the established cut-off values, cross-reactivity was observed in 8 out of 40 serum samples (20.0%) from clinically febrile patients without brucellosis when

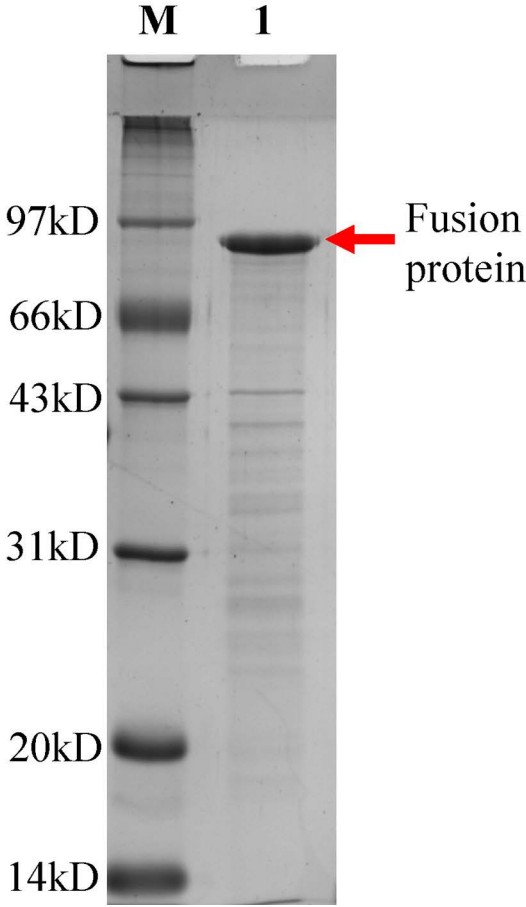

**Fig 1. SDS-PAGE (12%) analysis of the fusion protein expressed in prokaryotes.** M, the protein molecular weight marker. Lane 1, the purified fusion protein.

Table 2. **Comparison of serum OD450 results in different groups.**

| Groups | N | OD450 | $t$ value | $P$ |
|---|---|---|---|---|
| | | Mean ± SD | | |
| Fusion protein Positive | 100 | 0.6152 ± 0.2636[a] | 14.50 | <0.001 |
| Fusion protein Negative | 136 | 0.1989 ± 0.0998[b] | | |
| LPS Positive | 100 | 0.8610 ± 0.3417 | 22.17 | <0.001 |
| LPS Negative | 136 | 0.1759 ± 0.0247 | | |

[a]Compared with LPS Positive, $t$=5.695, $P$<0.001;

[b]Compared with LPS Negative, $t$=10.79, $P$<0.001.

tested with the fusion protein. Among these, there were 2 cases of *Staphylococcus aureus*, and 1 case each of *Pseudomonas putida*, *Moraxella osloensis*, *Rothia mucilaginos*, *Klebsiella pneumoniae*, *Pseudomonas aeruginosa*, and *Streptococcus dysgalactiae*. Cross-reactivity with LPS was observed in 16 cases (40.0%), including 8 cases of *Escherichia coli*, 3 cases of *Staphylococcus aureus*, and 1 case each of *Klebsiella pneumoniae*, *Moraxella osloensis*, *Enterococcus faecium*, *Moraxella osloensis*, and *Pseudomonas aeruginosa*. All data are detailed in S1 Table.

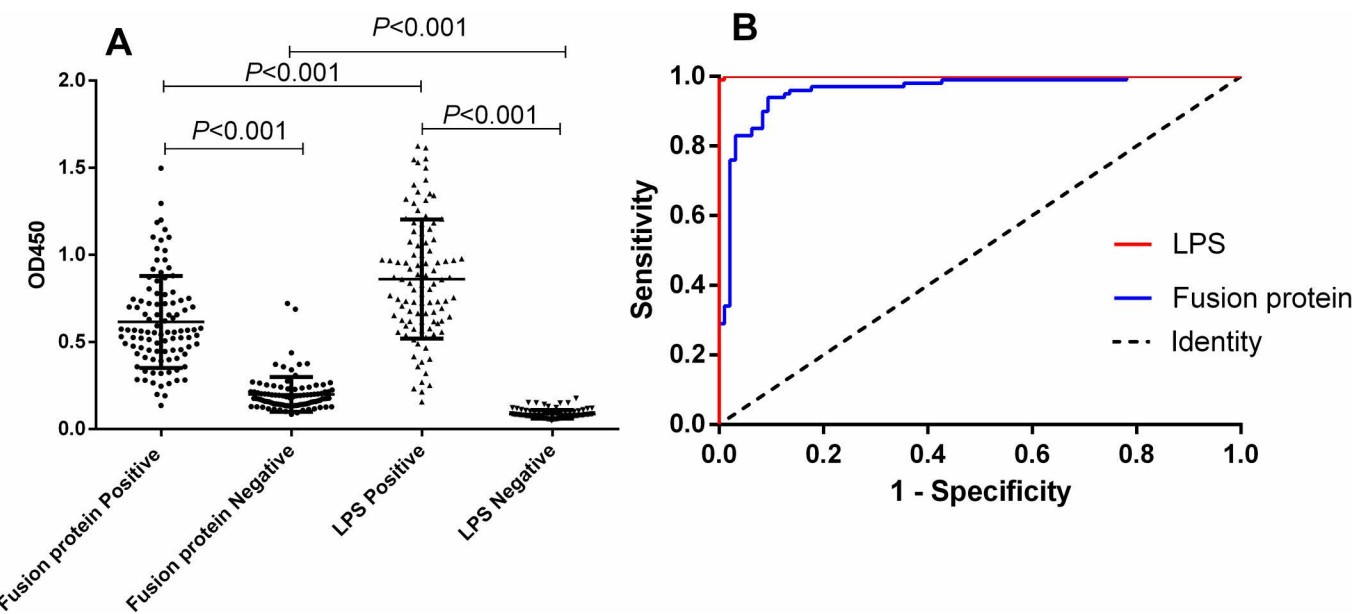

**Fig 2. I-ELISA detection of human serum samples.** A, the ROC curve for the human serum samples, illustrating the diagnostic sensitivity and specificity of the fusion protein-based assay. B, a dot plot representation of the OD450 values for both positive and negative human serum samples, with each dot representing an individual sample.

**Table 3. Positive predictive value and negative predictive value at different cut-off values.**

| Antigen | cut-off value | Positive | | Negative | | Accuracy (%) | PPV (%) | NPV (%) |
|---|---|---|---|---|---|---|---|---|
| | | TP | FN | TN | FP | | | |
| Fusion Protein | 0.2774 | 94 | 6 | 87 | 9 | 92.35 | 91.26 | 93.55 |
| LPS | 0.1953 | 99 | 1 | 96 | 0 | 99.49 | 100.0 | 98.97 |

TP, true positives; TN, true negatives; FP, false positives; FN, false negatives; Accuracy, (TP+TN/TP+FN+TN+FP) × 100; PPV, positive predictive value (TP/TP+FP) ×100; NPV, negative predictive value (TN/TN+FN) × 100.

## 4. Discussion

Brucellosis is a key zoonotic infectious disease prioritized for prevention and control in China. The disease is diverse, with complex pathogens and a wide geographic distribution. Humans are easily infected with *Brucella*, resulting in severe fever symptoms known as undulant fever, which, if misdiagnosed, can progress to a chronic condition accompanied by severe complications affecting the musculoskeletal, cardiovascular, and central nervous systems [25]. Therefore, establishing a rapid and accurate diagnostic method is crucial for the prevention and control of this disease.

Currently, most commercially available *Brucella* testing kits are based on the LPS present on the surface of *Brucella* [26]. Typically, obtaining LPS requires culturing live *Brucella* in high-level biosafety laboratories, which imposes significant demands on laboratory conditions [27]. Moreover, LPS shares some common antigens with *Escherichia coli* O:157, *Yersinia enterocolitica* O:9, *Salmonella* O:30, *Vibrio cholerae* O:1, and *Aeromonas hydrophila* [28], leading to potential cross-reactivity during testing and reduced diagnostic specificity. Therefore, finding alternative candidate antigens to LPS is critical for improving the serological diagnosis of brucellosis.

Extensive research indicates that *Brucella* outer membrane proteins (OMPs) possess strong immunogenicity and are potential new candidates for *Brucella* subunit vaccines and diagnostic antigens [29,30]. BP26 was initially identified as a major immunoreactive protein in animals infected with *Brucella*, showing good immunogenicity in cattle, sheep, goats, and humans [31], and is considered one of the most suitable *Brucella* vaccine markers. Although Omp10 does not produce serological reactions in naturally infected cattle, it does react in naturally infected sheep, making it a potential candidate not only for *Brucella* subunit vaccines but also for *Brucella* typing diagnostic antigens [32]. Research by Rezaei et al. suggests that Omp16 has good immunogenicity and can be a potential candidate for the development of oral and subunit vaccines for *Brucella* [33]. Sun et al. used recombinant capripoxvirus to express *Brucella* Omp25 in mice, demonstrating that Omp25 is a candidate antigen for subunit vaccines [34]. The Omp2 protein includes two highly homologous genes, Omp2a and Omp2b. Sha et al. used bioinformatics to analyze the linear B-cell epitopes of Omp2b, and the predictions of the Omp2b protein antigenic epitopes provide important theoretical basis for constructing ideal polyvalent *Brucella* antigenic epitope vaccines [35]. Omp31 is a surface porin protein of *Brucella*, and studies indicate that Omp31 may serve as a candidate for combined vaccines to enhance immunogenicity [36]. These antigenic proteins have some cross-reactivity with *Vibrio parahaemolyticus*, *Listeria monocytogenes*, *Legionella pneumophila*, *Salmonella* spp., and *Vibrio parahaemolyticus* but do not cross-react with *Yersinia enterocolitica* O9 and *Escherichia coli* O157:H7, this may be related to the fact that both *Yersinia enterocolitica* O9 and *Escherichia coli* O157:H7 contain LPS and share similar structures or epitopes with *Brucella*, whereas these proteins do not share epitopes with *Yersinia enterocolitica* O9 and *Escherichia coli* O157: H7 [37,38].

Therefore, this study predicted 31 epitopes from BP26, Omp10, Omp16, Omp25, Omp2a, Omp2b, and Omp31 and used fusion proteins as diagnostic antigens. Following the analysis of serum samples, it was observed that the OD450 values of positive serum for the detection of fusion proteins was lower than that of LPS. Conversely, the OD450 values of negative serum was found to be higher than that of LPS. This discrepancy may potentially compromise the accuracy of the results obtained from fusion protein detection. Comparison of the constructed fusion protein with LPS showed that the sensitivity of both diagnostics was comparable. The use of multiepitope fusion protein offers a significant advantage in terms of reduced cross-reactivity with other pathogens. This is particularly important given that traditional antigens, such as LPS, can exhibit cross-reactivity with antigens from other bacteria, leading to false positives and compromised diagnostic accuracy. By designing a fusion protein that targets specific epitopes of *Brucella* outer membrane proteins, we have been able to develop a diagnostic tool that is less likely to react with antigens from non-*Brucella* bacteria, thus enhancing the specificity of our assay. However, the cross-reactivity was still more severe, which may be related to the fact that the purification of the fusion protein was not good enough after our prokaryotic expression, which was only 90.1%, and the presence of heteroproteins of *Escherichia coli* in the fusion protein may have contributed to its cross-reactivity with sera infected with other pathogenic bacteria. Future studies should further purify the fusion protein to eliminate these possible interferences. The production of LPS requires culturing live *Brucella*, which poses significant biosafety risks and laboratory infrastructure demands. In contrast, our fusion protein can be expressed in a prokaryotic system, simplifying the production process and reducing associated costs. This approach not only saves time but also avoids the preparation of LPS antigens, making the diagnostic process safer and more convenient [39]. Moreover, the fusion protein provides a safer and more convenient alternative to LPS.

In summary, this study leverages the advantages of bioinformatics and fusion protein technology to successfully develop a novel diagnostic antigen for brucellosis serological testing.

This new antigen not only theoretically demonstrates superior recognition specificity and sensitivity but also provides new ideas and methods for the diagnosis of brucellosis. Nevertheless, our study is subject to several limitations. Firstly, the demographic information pertaining to the samples was ambiguous, and the selection criteria were restricted solely to serological positivity, thereby excluding culture-positive samples. Secondly, the sample size was constrained, being limited to a single geographic region. It is imperative to expand the sample collection to encompass a broader range of areas in order to validate the detection of this fusion protein, particularly by incorporating real-world random samples.

## Supporting information

**S1 Text. Protocol to SAT and prokaryotic expression of fusion protein.**
(DOCX)

**S1 Table. Sheet 1, OD450 of positive sera; sheet 2, OD450 of negative sera; sheet 3, data for cross-reactivity assessment.**
(XLSX)

## Acknowledgments

We thank the China Animal Health and Epidemiology Center for the gift of LPS, and the Xuzhou Center for Disease Control and Prevention for the gift of human brucellosis sera (positive and negative).

## Author contributions

**Conceptualization:** Meixue Yao, Dehui Yin.

**Data curation:** Yujia Xie, Liping Guo.

**Funding acquisition:** Dehui Yin.

**Methodology:** Yujia Xie, Liping Guo, Xinru Qi, Shiqi Zhao, Qichuan Pei, Yixiao Chen, Qi Wu.

**Writing – original draft:** Yujia Xie.

**Writing – review & editing:** Meixue Yao, Dehui Yin.

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
