## [Decision Letter · Decision Letter 0]

23 Dec 2024

PNTD-D-24-01432

Establishment of an I-ELISA Method Based on Multi-Epitope Fusion Protein for Diagnosis of Human Brucellosis

Dear Dr. Yin,

Thank you for submitting your manuscript to PLOS Neglected Tropical Diseases. After careful consideration, we feel that it has merit but does not fully meet PLOS Neglected Tropical Diseases's publication criteria as it currently stands. Therefore, we invite you to submit a revised version of the manuscript that addresses the points raised during the review process.

Please submit your revised manuscript within 60 days Feb 21 2025 11:59PM. If you will need more time than this to complete your revisions, please reply to this message or contact the journal office at plosntds@plos.org. Please include the following items when submitting your revised manuscript:

We look forward to receiving your revised manuscript.

Kind regards,

Mathieu Picardeau

Section Editor

Shaden Kamhawi

co-Editor-in-Chief

Paul Brindley

co-Editor-in-Chief

**Journal Requirements:**

1) Thank you for including an Ethics Statement for your study. Please include:

i) A statement that formal consent was obtained (must state whether verbal/written) OR the reason consent was not obtained (e.g. anonymity). NOTE: If child participants, the statement must declare that formal consent was obtained from the parent/guardian.].

2) Please amend your detailed Financial Disclosure statement. This is published with the article. It must therefore be completed in full sentences and contain the exact wording you wish to be published.

**Comments to the Authors:**

Please note that one of the reviews is uploaded as an attachment.

**Reviewers' Comments:**

Reviewer's Responses to Questions

**Key Review Criteria Required for Acceptance?**

**Methods**

-Are the objectives of the study clearly articulated with a clear testable hypothesis stated?

-Is the study design appropriate to address the stated objectives?

-Is the population clearly described and appropriate for the hypothesis being tested?

-Is the sample size sufficient to ensure adequate power to address the hypothesis being tested?

-Were correct statistical analysis used to support conclusions?

-Are there concerns about ethical or regulatory requirements being met?

Reviewer #1: - Are the study objectives clearly articulated and a clear testable hypothesis established?

Yes, the authors clearly explain their objective.

- Is the study design adequate to address the stated objectives?

The design is correct and the number of samples is adequate.

- Is the population clearly described and adequate for the hypothesis being tested?

I believe that the authors should better explain the characteristics of the samples (the dates on which they were collected, the tests that were performed, ...).

The authors report that the samples are positive for the agglutination test but the gold standard for brucellosis is culture, the authors should explain why they used a serological test as a confirmatory test to evaluate another serological test.

- Is the sample size sufficient to ensure adequate power to address the hypothesis being tested?

Yes,

- Were correct statistical analyses used to support the conclusions?

I felt a lack of statistical comparison between the LPS and fusion protein results.

- Are there concerns about compliance with ethical or regulatory requirements?

No, all ethical requirements were met.

Reviewer #2: Accept with minor revisions as detailed in attachment for clarity.

Reviewer #3: methods do not state how OD values were used for the "ROC curve analysis" and area under the curve analysis. This undermines the statistical analysis and interpretation for their main findings for iELISA specificity and accuracy conclusions.

Source of LPS used as control is not stated.

**Results**

-Does the analysis presented match the analysis plan?

-Are the results clearly and completely presented?

-Are the figures (Tables, Images) of sufficient quality for clarity?

Reviewer #1: -Does the analysis presented match the analysis plan?

The results lack a statistical complement to compare the obtained results.

-Are the results presented clearly and completely?

Yes, the results are clear but I feel a lack of more statistical comparisons between the two groups evaluated.

-Are the figures (tables, images) of sufficient quality to be clear?

Yes,

Reviewer #2: Minor revisions: The authors should include results of the independent t-tests outlined in the methods and presently only presented in Figure 2B.

Reviewer #3: Unable to safely interpret data without explaining how the OD readings go into the diagnostic curve (AUC) and ultimately into the values for prediction (TABLE 2).

I do not understand how data goes into creating Figure 1 panel A.

It would also appear that the LPS only has significant activity as bait for their iELISA, if not more so than their fusion protein.

**Conclusions**

-Are the conclusions supported by the data presented?

-Are the limitations of analysis clearly described?

-Do the authors discuss how these data can be helpful to advance our understanding of the topic under study?

-Is public health relevance addressed?

Reviewer #1: -Are the conclusions supported by the data presented?

Yes, the conclusions are clear and supported by the results.

-Are the limitations of analysis clearly described?

I did not find a paragraph explaining the limitations of the study. The authors could add a paragraph explaining the limitations of the study.

-Do the authors discuss how these data can be helpful to advance our understanding of the topic under study?

The authors need to go more into the benefits of using fusion protein, I feel they can better explain the benefits of low cross-reactivity.

-Is public health relevance addressed?

Yes, the authors explain well the importance of looking for other diagnostic options for brucellosis.

Reviewer #2: Accept, with minor revisions as suggested in attachment.

Reviewer #3: Unable to interpret accuracy of the conclusion without logic behind calculations.

**Editorial and Data Presentation Modifications?**

Reviewer #1: -Line 39: write what LPS stands for

-Line 116 - 123: write the full name of the acronym

-Line 139: Explain the characteristics of the samples better (on what dates were they collected, tests that were performed to confirm the disease, ...)

-Line 140: Do the authors consider it appropriate that the confirmatory method for positive samples be agglutination? Since a serological method is being evaluated and the confirmatory test for positive samples is another serological method, other confirmatory tests are needed for positive samples.

-Line 274: Perform statistical tests to compare the two groups and then discuss those results.

-Line 296 - 300: the authors explain the cross-reactivity with 7 pathogens but in the discussion they do not explain why this could occur. Does the fusion protein sequence have similarity with the sequence of these pathogens?

Did the authors perform any analysis of the sequences?

-In the discussion I consider it important to include a paragraph about the limitations of the study.

Reviewer #2: Minor revisions- see comments by line number in attachment.

Reviewer #3: tense of methods section detailing cloning of fusion construct is written as a protocol (do this then do that).

figure legends lack any informative detail.

**Summary and General Comments**

Reviewer #1: (No Response)

Reviewer #2: I think this was a thoughtful, well executed study that draws appropriate conclusions from the results showing that the designed fusion protein can greatly reduce cross-reactivity (false positive rate) compared to the current method using LPS for serological testing. All suggested revisions are fairly minor but meant to tighten up the manuscript and perhaps help make the findings more compelling. I do think it would be interesting to rerun the ROC curve analyses grouping the Brucella-negative sera and sera used to compare cross-reactivity with other infections to consider how the fusion protein compares to LPS under "real world" conditions when disease status is not known prior. Otherwise, I think this is an important study with immediate relevancy and public health application.

Reviewer #3: methods do not state how OD values were used for the "ROC curve analysis" and area under the curve analysis. This undermines the statistical analysis and interpretation for their main findings for iELISA specificity and accuracy conclusions.

Source of LPS used as control is not stated.

Unable to interpret accuracy of the conclusion without logic behind calculations.

PLOS authors have the option to publish the peer review history of their article (what does this mean? ). If published, this will include your full peer review and any attached files.

**Do you want your identity to be public for this peer review?** For information about this choice, including consent withdrawal, please see our Privacy Policy .

Reviewer #1: No

Reviewer #2: No

Reviewer #3: No

**Figure resubmission:**
---

## [Decision Letter · Decision Letter 1]

18 Mar 2025

Dear Dr Yin,

We are pleased to inform you that your manuscript 'Establishment of an I-ELISA Method Based on Multi-Epitope Fusion Protein for Diagnosis of Human Brucellosis' has been provisionally accepted for publication in PLOS Neglected Tropical Diseases.

Best regards,

Mathieu Picardeau

Section Editor

Mathieu Picardeau

Section Editor

Shaden Kamhawi

co-Editor-in-Chief

Paul Brindley

co-Editor-in-Chief

Reviewer's Responses to Questions

**Key Review Criteria Required for Acceptance?**

**Methods**

-Are the objectives of the study clearly articulated with a clear testable hypothesis stated?

-Is the study design appropriate to address the stated objectives?

-Is the population clearly described and appropriate for the hypothesis being tested?

-Is the sample size sufficient to ensure adequate power to address the hypothesis being tested?

-Were correct statistical analysis used to support conclusions?

-Are there concerns about ethical or regulatory requirements being met?

Reviewer #1: (No Response)

**Results**

-Does the analysis presented match the analysis plan?

-Are the results clearly and completely presented?

-Are the figures (Tables, Images) of sufficient quality for clarity?

Reviewer #1: (No Response)

**Conclusions**

-Are the conclusions supported by the data presented?

-Are the limitations of analysis clearly described?

-Do the authors discuss how these data can be helpful to advance our understanding of the topic under study?

-Is public health relevance addressed?

Reviewer #1: (No Response)

**Editorial and Data Presentation Modifications?**

Reviewer #1: (No Response)

**Summary and General Comments**

Reviewer #1: (No Response)

PLOS authors have the option to publish the peer review history of their article (what does this mean? ). If published, this will include your full peer review and any attached files.

**Do you want your identity to be public for this peer review?** For information about this choice, including consent withdrawal, please see our Privacy Policy .

Reviewer #1: No

---

## [Editor Report · Acceptance letter]

Dear Dr Yin,

We are delighted to inform you that your manuscript, "Establishment of an I-ELISA Method Based on Multi-Epitope Fusion Protein for Diagnosis of Human Brucellosis," has been formally accepted for publication in PLOS Neglected Tropical Diseases.

Best regards,

Shaden Kamhawi

co-Editor-in-Chief

Paul Brindley

co-Editor-in-Chief
